# Design and DOF Analysis of a Novel Compliant Parallel Mechanism for Large Load

**DOI:** 10.3390/s19040828

**Published:** 2019-02-17

**Authors:** Xiaochuan Wu, Yi Lu, Xuechao Duan, Dan Zhang, Wenyao Deng

**Affiliations:** 1Key Laboratory of Electronic Equipment Structure Design, Ministry of Education of China, Xidian University, Xi’an 710071, China; xcwu@stu.xidian.edu.cn (X.W.); dan_zhang99@hotmail.com (D.Z.); wydeng@stu.xidian.edu.cn (W.D.); 2College of Information Science and Technology, Shihezi University, North 4th Road, Shihezi 832003, China; luyi@xidian.edu.cn

**Keywords:** compliant mechanism, flexural joint, screw theory, symbolic formulation

## Abstract

The degree of freedom (DOF) and motion characteristics of a kind of compliant spherical joint were analyzed based on the screw theory, and a new design scheme for force-inversion of the compliant spherical joint was proposed in this paper. A novel type of six DOF compliant parallel mechanism (CPM) was designed based on this scheme to provide a large load capacity and achieve micrometer-level positioning accuracy. The compliance matrix of the new type of CPM was obtained through matrix transformation and was then decomposed into its generalized eigenvalues. Then, the DOF of the mechanism was numerically analyzed based on the symbolic formulation. The finite element analysis model of the compliant parallel mechanism was established. The static load analysis was used to verify the large load capacity of the mobile platform. By comparing the deformation obtained by the compliance matrix numerical method with the deformation obtained by the finite element method, the correctness of the compliance matrix and the number of the DOF of the CPM was verified.

## 1. Introduction

The motion of a complaint mechanism is caused by the elastic deformation of flexure elements when bearing loads [1,2,3,4]. Compared with the traditional rigid-body mechanism, it has the advantages of no friction, no lubrication required, compact structure, integrated molding, no assembly and high precision. Complaint mechanisms have been widely used in various precision instruments [5,6] including nanomanipulator [7], micropositioner [8], optical scanning mirrors [9], scanning transmission X-ray microscopy [10] and sensors [11], etc. However, the compliant mechanism usually fails to bear large loads. In the current research, there are few published results taking the load into account. In the process of judging the DOF of the compliant mechanism, the boundary between the DOF and the constraint is different from that of the rigid member, which can be directly calculated by the formula of the DOF; therefore, the DOF determination of the compliant mechanism is challenging in the analysis and synthesis for compliant mechanisms.

The DOF of a compliant mechanism is decided by geometric parameters, material properties and load. Currently, there is less research on the load capacity of the compliant mechanism, and the compliant mechanism is weak in bearing large loads due to the structural characteristics of the compliant hinge in the mechanism. Shi et al. presented workspace optimization of MEMS flexure-based hexapod nanopositioner previously built by the National Institute of Standards and Technology (NIST) and presented an analytical formulation and a search algorithm to determine the workspace of the flexure-based parallel mechanisms. A novel adaptive genetic algorithm was developed to conduct the single and bi-objective optimization for maximum translational and rotational workspace [12]; Shi et al. covered the kinematic modeling of a flexure-based, hexapod nanopositioner and calculated the actuation data for a set of commands for decoupled and coupled motions and obtained the Jacobian matrix of the mobile platform for the controller to calibrate the precision of the nanopositioner; however, it can hardly bear a considerable load [13]. Brouwer presented a precision MEMS-based six degrees-of-freedom (DOFs) manipulator. The purpose of the manipulator is to position a small sample (10 μm × 20 μm × 0.2 μm) in a transmission electron microscope. A parallel kinematic mechanism with slanted leaf-springs was used to convert the motion of six in-plane electrostatic comb-drives into six DOFs at the end-effector, which can only carry a small sample of 10~50 micrograms [14,15].

At the beginning of the design, it is necessary to analyze the motion characteristics, in which the DOF analysis is of the most importance. The DOF of a traditional rigid-body mechanism can be directly calculated by Grubler–Kutzbach’s formula, but this fails to judge the DOF of compliant mechanisms due to the flexure joints. Howell et al. first proposed the pseudo-rigid-body model method to analyze the DOF of a planar compliant mechanism [16]. Deshmukh et al. used a pseudo-rigid-body model (PRBM) method to design a flexure-based compliant parallel (4-bar) mechanism for a linear translational motion actuated via a precision slide, compared the results with those from finite element analysis, and verified the PRBM theory through experiments [17]. Su analyzed the motion characteristics of the general complaint mechanism based on screw theory and provided an important method for guiding the qualitative design of flexure mechanisms [18,19,20]. However, these methods proposed by Su are qualitative in process of judging the DOF and do not consider the influence of the size factor on the number of DOF. Li et al. defined a new generalized mechanism and derived the three main formulas for calculating the maximum and minimum DOF with screw theory and verified the factors affecting the DOF [21].

Based on the screw theory, the degree of freedom of the compliant spherical joint was firstly analyzed in this paper. A novel six-DOF CPM able to bear significant loads based on the compliant spherical joint was designed [22,23,24,25]. This type of mechanism is a mechatronic device with six DOFs and high-precision positioning in the space, which enables the mechanism to accurately detect the position of six DOFs in space after each branch installation is picked up. Thus the mechanism is widely applied in sensors [26], assembly of MEMS devices and micro-systems, and precision measurement of micro-systems. Instead of the present pseudo-rigid-body model method [27] and the DOF criteria based on screw theory [28], the method of the eigenwrench and the eigentwist decomposition in the mechanism compliance matrix is used to judge the DOF of the compliant mechanism. Firstly, by using the matrix transformation, the compliance matrix of the whole mechanism is established [29,30,31]. Then, the DOF of the mechanism is numerically analyzed by solving the generalized eigenvalues and eigenvectors of the compliance matrix. Finally, the compliance matrix of the parallel mechanism and the number of DOF were validated by comparing the deformation obtained by the compliance matrix numerical method with that from finite element simulation.

## 2. Analysis of the Compliant Spherical Joint Based on Screw Theory

### 2.1. Compliance Matrix of an RCCS (Right Circular Cross-Section) Flexural Joint

Under the condition that the ratio of the length to the radius of the elongated cylindrical rod is less than five, the slender rod joint has a total of three DOFs, which are rotated by the three axes of the coordinate system ($\theta_{x},\theta_{y}$ and $\theta_{z}$).

An RCCS flexural joint [32,33,34,35,36] is shown in Figure 1a. The ratio of the minimum cross-sectional diameter *t*-min of the compliant joint to the maximum cross-sectional diameter *t*-max is less than or equal to 1/10, and the distance of the joint between ends is 2r. The joint can achieve three DOFs rotation in the three-dimensional space shown in Figure 1a, namely bending along the *y*-axis and *z*-axis and axial torsion along the *x*-axis. This type of flexural joint has higher precision than the slender compliant spherical joint [37].

The two rods in Figure 1 are the two ends of the two different parts connected to it, because the deformation of the two parts is negligible relative to the compliant joint, so they are not taken into account in the numerical analysis. In the finite element analysis, in view of the actual force of the RCCS compliant spherical joint in the single branch of the mechanism, the boundary condition is gravity, the fixed constraint is applied at one end of the compliant spherical joint, and the other end of the round end face is applied with a direction-constant force or a constant amount of torque.

When the finite element analysis software is used for the RCCS compliant spherical joint, the linear analysis module is adopted. The reason for not using a non-linear analysis module is the verification of compliance matrix of the mechanism is compared and analyzed by the numerical method based on the compliance matrix and deformation obtained by the finite element method. The force matrix added during the compliance matrix numerical method is added in one step, and the compliance matrix is not changed in the process (even if the force matrix is applied in several steps, the compliance matrix does not change for the numerical method). For the finite element method, the compliance/stiffness matrix is dynamically changing after each loading of the generalized force. Therefore, in the process of finite element analysis, the force matrix is not used in several steps in the nonlinear analysis module. The material of the mechanism is 60 steel of structural steel, and the elastic modulus of the material is *E* = 200 GPa. Poissons’ ratio is ʋ = 0.3, density is 7.85 g/cm^3^; element type is solid186; automatic meshing is adopted; and mesh density Relevance = 20. After mesh, the number of nodes is 7281 and the number of elements is 3588. As for mesh quality, the element quality average is 0.61, the aspect ratio average is 3.19 and the Jacobian ratio average is 1.02; the boundary conditions include a fixed constraint applied on the bottom surface of the base, gravity, a constant magnitude force along the *x*-axis applied at the center point of the mobile platform, and the simulation analysis is completed in one step; after finite element analysis, the compliant spherical joint has three DOFs, which are rotations around the three coordinate axes (*θ_x_*, *θ_y_*, and *θ_z_*). The angle of rotation of the RCCS compliant spherical joint around the *x*, *z*, and *y*-axes is −0.88°~+ 0.88°, −0.88°~+0.88° and −0.57°~+0.57°, respectively.

According to the mechanics of elasticity, the compliance matrix of an RCCS flexural joint can be written as [38]:(1)$$C_{\mathrm{hinge}}=\left(\begin{array}{cccccc} 0 & 0 & 0 & c_{\theta_{x}-M_{x}} & 0 & 0\\ 0 & 0 & c_{\theta_{y}-F_{x}} & 0 & c_{\theta_{y}-M_{y}} & 0\\ 0 & c_{\theta_{z}-F_{y}} & 0 & 0 & 0 & c_{\theta_{z}-M_{z}}\\ c_{x-F_{y}} & 0 & 0 & 0 & 0 & 0\\ 0 & c_{y-F_{y}} & 0 & 0 & 0 & c_{y-M_{z}}\\ 0 & 0 & c_{z-F_{y}} & 0 & c_{z-M_{y}} & 0 \end{array}\right)$$
where $c_{\theta_{x}-M_{x}}=\frac{32}{\pi G}\left(\int_{0}^{2r}\frac{1}{t{\left(x\right)}^{4}}dx\right)$, $c_{\theta_{y}-M_{y}}=c_{\theta_{z}-M_{z}}=\frac{64}{\pi E}\left(\int_{0}^{2r}\frac{1}{t{\left(x\right)}^{4}}dx\right)$, $c_{x-F_{y}}=\frac{4}{\pi E}\left(\int_{0}^{2r}\frac{1}{t{\left(x\right)}^{2}}dx\right)$, $c_{y-F_{y}}=c_{z-F_{z}}=\frac{64}{\pi E}\left(\int_{0}^{2r}\frac{x^{2}}{t{\left(x\right)}^{4}}dx\right)+\frac{7E}{6G}c_{x-F_{y}}$, $c_{y-M_{z}}=-c_{z-M_{y}}=\frac{64}{\pi E}\left(\int_{0}^{2r}\frac{x}{t{\left(x\right)}^{4}}dx\right)$, $c_{y-M_{z}}=c_{\theta_{z}-F_{y}},$
and $c_{z-M_{y}}=c_{\theta_{y}-F_{z}}.$
where *E* is the Youngs modulus and *G* is the shear modulus. t(x) can be expressed as:(2)$$\left\{ \begin{array}{l} t(x)=t+2\left[r-\sqrt{r^{2}-{\left(r-x\right)}^{2}}\right],x\in (0,r]\\ t(x)=t+2\left[r-\sqrt{r^{2}-{\left(x-r\right)}^{2}}\right],x\in \left(r,2r\right] \end{array}\right.$$

Then, the corresponding material parameters are substituted into Equation (1) to obtain a three DOFs RCCS flexural joint with a 6×6 compliance matrix.

### 2.2. The Compliance Matrix of RCCS Serial Chain

As shown in Figure 2, one end of an RCCS flexural joint was fixed as the reference object 1, and the other end of the RCCS flexural joint can be regarded as the mobile body 3. A serial compliant mechanism consisting of two RCCS flexural joints connected through the rigid body 2 can be obtained [39,40]. The parts 1, 2 and 3 are rigid bodies in this case.

$\left[C_{i}\right]$ denotes the compliance matrix of the *i*-th flexure member. Within the elastic limit range, the deformation of a serial chain is a superposition of each flexure element’s deformation in the same coordinate system. For a serial mechanism, the relationship between the overall compliance matrix and other flexure member compliance matrices can be expressed as follows:(3)$$\left[C\right]=\sum_{i=1}^{n}\left[Ad_{i}\right]\left[C_{i}\right]{\left[Ad_{i}\right]}^{-1}$$
where $\left[Ad_{i}\right]$ is the transformation matrix transforming the compliance matrix of the *i*-th flexure element to the functional body coordinate system.

As shown in Figure 2, *o*-*xyz* is considered as the global coordinate system; the coordinate system *o*-*x*_2_*y*_2_*z*_2_ is fixed to the moving body. The compliance matrix of the flexure element under the coordinate systems *o*-*x*_1_*y*_1_*z*_1_ and *o*-*x*_2_*y*_2_*z*_2_ can be converted to the coordinate system *o*-*xyz* through the transformation matrices $\left[Ad_{1}\right]$ and $\left[Ad_{2}\right]$. Where $R_{1}=R_{2}=I$, I is a 3 × 3 unit matrix, $d_{1}={\left(\begin{array}{ccc} -l, & 0, & 0 \end{array}\right)}^{T}$, $d_{2}={\left(\begin{array}{ccc} 0, & 0, & 0 \end{array}\right)}^{T}$. $\left[Ad_{1}\right]$ and $\left[Ad_{2}\right]$ can be written as:(4)$$\begin{array}{cc} \left[Ad_{1}\right]=\left[\begin{array}{cc} R_{1} & 0\\ D_{1}R_{1} & R_{1} \end{array}\right], & \left[Ad_{2}\right]=\left[\begin{array}{cc} R_{2} & 0\\ D_{2}R_{2} & R_{2} \end{array}\right] \end{array}$$

According to Equation (3), the compliance matrix of a serial chain based on compliant joints can be expressed as:(5)$$C_{\mathrm{leg}}=\left[Ad_{1}\right]\left[C_{1}\right]{\left[Ad_{1}\right]}^{-1}+\left[Ad_{2}\right]\left[C_{2}\right]{\left[Ad_{2}\right]}^{-1}$$

### 2.3. A combined Design of Serial Compliant Joints with a Function of Interchanging Press and Pull

When a slender rod is subjected to pressure, it exhibits a property completely different from the strength failure. As shown in Figure 3, the lower end of the slender rod is fixed, and the upper end is free. When the pressure is beyond the limit value of $F_{cr}$, the slender bar can return to its original state after the force is removed. When the pressure *F* gradually exceeds this limit value, the slender bar will maintain the balance of the curve shape and cannot restore the original shape. The slender bar loses its balance of a linear shape and transitions to a curve balance called buckling phenomenon [41,42,43].

As shown in Figure 3b, the RCCS flexural joint is similar to the compression bar and tends to exhibit the above-mentioned buckling phenomenon. After the compression bar is destabilized, a small increase in pressure will cause a significant increase in bending deformation, at which point the compression bar has lost its ability to carry the load. Instability causes the failure of the compression bar, which can cause damage to the entire machine or the corresponding structure, resulting in irreparable damage.

To avoid the damage of the mechanism caused by the instability of the RCCS flexural joint due to excessive pressure, this paper adopted a new structure to change the pressure on the RCCS flexural joint to the tension shown in Figure 4a,b.

A novel serial RCCS flexural joint chain structure based on an RCCS flexural joint is illustrated in Figure 4a. Part 1 is the fixed end; Part 2 is the rigid rod connecting two RCCS flexural joints. Part 3 is the free end bearing various forces or torque; and Part 4 is the RCCS flexural joint. As shown in Figure 4b, when the two U-shaped free ends are subjected to the axial pressure *F*/2 along the coordinate system, the two ends of the RCCS flexural joint will be subjected to two forces of the same value *F* and opposite directions. In this way, the pressure applied to the RCCS flexural joint is converted into tension, so that the structure can bear a much larger axial pressure without an instability problem.

The new serial RCCS flexural joint chain compliance matrix that can bear a large pressure load is solved in the same way as the previous compliance matrix solution. According to Equations (1)–(4), substituting *l* = 230 mm into the Equation (5) obtains a new serial RCCS flexural joints branch compliance matrix $C_{\mathrm{newleg}}$:(6)$$C_{\mathrm{newleg}}=\left[Ad_{1}\right]\left[C_{1}\right]{\left[Ad_{1}\right]}^{-1}+\left[Ad_{2}\right]\left[C_{2}\right]{\left[Ad_{2}\right]}^{-1}$$

### 2.4. Compliance Matrix of CPM Based on the New Serial RCCS Flexural Joint Branch Chain

As shown in Figure 5a, the CPM in this paper is mainly composed of six new serial RCCS flexural joint branch chains as shown in Figure 4a. The angle between the center of the adjacent compliant spherical joint in each of branches near the mobile platform is 60° and the angle between the single and vertical branch is *α* = 12° [44].

From Equation (6), the compliance matrix of a single new serial RCCS flexural joint branch can be calculated. According to the relationship between the compliance matrix and stiffness matrix $K_{\mathrm{newleg}}=C_{\mathrm{newleg}}^{-1}$, the stiffness matrix of a new serial RCCS flexural joint can be written as:(7)$$K_{\mathrm{newleg}}=C_{\mathrm{newleg}}^{-1}$$

To facilitate the study of the motion of the platform on the entire CPM and the deformation and displacement of the mobile platform during the stress process, it is necessary to transform the stiffness matrix of each branch into the fixed coordinate system of the mobile platform. As shown in Figure 5b, the single-branched coordinate system *o*-*x*_1_*y*_1_*z*_1_ needs to be matrix transformed into the upper platform fixed coordinate system *o*-*xyz*.

The coordinate transformation matrix of each single-branched local coordinate system *o_i_*-*x_i_**y_i_**z_i_* (*i* = 1, 2, …, 6) to the coordinate system *o*-*xyz* can be expressed as:(8)$$\begin{array}{cc} R_{i}=R_{x}\left(\alpha_{i}\right)R_{y}\left(\beta_{i}\right)R_{z}\left(\gamma_{i}\right), & i=1,2,\ldots ,6 \end{array}$$
(9)$$\begin{array}{cc} d_{i}={\left(r_{1}c\rho_{i},-h,r_{1}c\zeta_{i}\right)}^{T}, & i=1,2,\ldots ,6 \end{array}$$
where $c\rho =\cos(\rho_{i}),\;s\zeta =\sin(\zeta_{i})$, *ρ* and *ζ* respectively indicate the angle between the line which is composed of the origin of each local coordinate system projected on the *xz* plane of the global coordinate system; the origin of the global coordinate system; and the *x*-axis and *z*-axis of the global coordinate system; $R_{x}\left(\alpha_{i}\right)$, $R_{y}\left(\beta_{i}\right)$ and $R_{z}\left(\gamma_{i}\right)$ represent the transformation matrix around the *x*-axis, *y*-axis and *z*-axis, respectively. They can be written as:(10)$$\begin{array}{ccc} R_{x}\left(\alpha_{i}\right)=\left[\begin{array}{ccc} 1 & 0 & 0\\ 0 & c\alpha_{i} & -s\alpha_{i}\\ 0 & s\alpha_{i} & c\alpha_{i} \end{array}\right], & R_{y}\left(\beta_{i}\right)=\left[\begin{array}{ccc} c\beta_{i} & 0 & s\beta_{i}\\ 0 & 1 & 0\\ -s\beta_{i} & 0 & c\beta_{i} \end{array}\right], & R_{z}\left(\gamma_{i}\right)=\left[\begin{array}{ccc} c\gamma_{i} & -s\gamma_{i} & 0\\ s\gamma_{i} & c\gamma_{i} & 0\\ 0 & 0 & 1 \end{array}\right] \end{array}$$
where $c\alpha_{i}=\cos(\alpha_{i}),\;s\alpha_{i}=\sin(\alpha_{i})$, $c\beta_{i}=\cos(\beta_{i}),\;s\beta_{i}=\sin(\beta_{i})$, $c\gamma_{i}=\cos(\gamma_{i}),\;s\gamma_{i}=\sin(\gamma_{i})$.

The geometric parameters of the CPM and the corresponding material property parameters are shown in Table 1. The Z-Y-X Euler angle coordinate transformation is applied. *γ_i_* represents the angle at which the *i*-th branch rotates about the *z*-axis in the local coordinate system; *β_i_* represents the angle at which the *i*-th branch rotates about the *y*-axis in the local coordinate system; *α_i_* represents the angle at which the *i*-th branch rotates about the *x*-axis in the local coordinate system, *i* = 1, 2, …, 6. $r_{1}$ represents the radius of the dotted circle consisting of the each single-branch local coordinate system origin as shown in Figure 5c. The material and geometric parameters are designed according to the pre-planning according to the medium-scale prototype, and the material cost and fabricating cost are taken into account. The compliant spherical joint symmetry center section’s size of the circle of 2 mm is much smaller than the size of other components in the mechanism. The reason why it does not choose a smaller size is also based on the consideration of whether it can withstand large loads and the results of finite element simulation.

According to Equation (7), the stiffness matrix of the large load CPM can be calculated as:(11)$$\begin{array}{cc} & K_{i}=K_{\mathrm{newleg}}\\ K & =\sum_{i=1}^{6}\left[Ad_{i}\right]\left[K_{i}\right]{\left[Ad_{i}\right]}^{-1}\\ & =\left(\begin{array}{cccccc} 0 & 0 & -2.7298\times 10^{6} & 9.2440\times 10^{3} & 0 & 0\\ 0 & 0 & 0 & 0 & 4.0877\times 10^{5} & 9.2440\times 10^{3}\\ 2.7298\times 10^{6} & 0 & 0 & 0 & 0 & 2.8298\times 10^{6}\\ 8.0849\times 10^{8} & 0 & 0 & 0 & 0 & 0\\ 0 & 2.1867\times 10^{5} & 0 & 0 & 0 & 0\\ 0 & 0 & 8.0849\times 10^{8} & -2.7298\times 10^{6} & 0 & 0 \end{array}\right) \end{array}$$

The compliance matrix of the large load CPM can be written as:(12)$$C=K^{-1}=\left(\begin{array}{cccccc} 0 & 0 & -1.2621\times 10^{-4} & 4.2738\times 10^{-7} & 0 & 0\\ 0 & 0 & 0 & 0 & 4.5732\times 10^{-6} & 0\\ 1.2621\times 10^{-4} & 0 & 0 & 0 & 0 & 4.2738\times 10^{-7}\\ 0.0374 & 0 & 0 & 0 & 0 & 1.2621\times 10^{-4}\\ 0 & 2.4464\times 10^{-6} & 0 & 0 & 0 & 0\\ 0 & 0 & 0.0374 & -1.2621\times 10^{-4} & 0 & 0 \end{array}\right)$$

## 3. The DOF Analysis of CPM Based on Compliant Branch

The compliant mechanism usually consists of a series of compliant components. In the process of the DOF analysis, the difference between DOFs and constraints are indiscernible. It is different from rigid members which can be directly calculated using the analytical formula of DOFs. The DOF determination of the compliant mechanism is of difficulty in the actual process. In this paper, the eigenwrench and eigentwist decomposition method in the mechanism compliance matrix were mainly used to judge the DOF of the compliance mechanism.

Because the deformation of the compliant mechanism satisfies the small deformation hypothesis according to elastic theory, the relationship between the wrench $\hat{W}$ and twist $\hat{T}$ can be expressed as:(13)$$\begin{array}{cc} \hat{T}=C\hat{W}, & \hat{W}=K\hat{T} \end{array}$$
(14)$$\hat{T}={\left(\begin{array}{cc} \theta^{T} & \delta^{T} \end{array}\right)}^{T}={\left(\begin{array}{cccccc} \theta_{x} & \theta_{y} & \theta_{z} & \delta_{x} & \delta_{y} & \delta_{z} \end{array}\right)}^{T}$$
(15)$$\hat{W}={\left(\begin{array}{cc} f^{T} & m^{T} \end{array}\right)}^{T}={\left(\begin{array}{cccccc} f_{x} & f_{y} & f_{z} & m_{x} & m_{y} & m_{z} \end{array}\right)}^{T}$$
where θ, δ, m, f are all composed of 3×1 vector elements, which represent rotational deformation, moving deformation, force, and torque respectively.

The compliance matrix ***C*** and stiffness matrix ***K*** are both 6×6 square arrays. If the stiffness matrix ***K*** is nonsingular, the compliance matrix ***C*** can be written as $C=K^{-1}$.

According to the calculation of the reciprocal product, the wrench $\hat{W}$ work was done along the twist $\hat{T}$, which can be written as:(16)$$\hat{T}\circ \hat{W}=\hat{T}\Delta \hat{W}=f^{T}\cdot \delta +m^{T}\cdot \theta$$
where ∘ is the reciprocal product of the wrench and twist, and ∆ is the swap operator defined as:
$$\Delta =\left(\begin{array}{cc} 0_{3\times 3} & I_{3\times 3}\\ I_{3\times 3} & 0_{3\times 3} \end{array}\right)$$

In a coordinate system Q, the wrench and twist are represented by $\hat{W}$ and $\hat{T}$, respectively. In the new coordinate system $Q^{\prime }$, the wrench and twist are represented by ${\hat{W}}^{\prime }$ and ${\hat{T}}^{\prime }$, respectively. The relationship between them can be expressed as:(17)$$\begin{array}{cc} {\hat{T}}^{\prime }{=\mathrm{Ad}}_{g}\hat{T}, & {\hat{W}}^{\prime }={\mathrm{Ad}}_{g}\hat{W} \end{array}$$
where ${\mathrm{Ad}}_{g}$ is the adjoint transformation matrix:(18)$${\mathrm{Ad}}_{g}=\left(\begin{array}{cc} R & 0\\ DR & R \end{array}\right)$$

As shown in Equation (18), the matrix R represents the 3×3 rotation matrix of the new coordinate system $Q^{\prime }$ with respect to the coordinate system Q, ***D*** is the skew-symmetric matrix defined by the translational vector ***d***, and 0 represents the 3×3 zero matrix.

It can be seen from the above equation that the compliance matrix and stiffness matrix in the new coordinate system can be obtained by the following equation
(19)$$\begin{array}{cc} C^{\prime }{=\mathrm{Ad}}_{g}C{\mathrm{Ad}}_{g}{}^{-1}, & W^{\prime }{=\mathrm{Ad}}_{g}W{\mathrm{Ad}}_{g}{}^{-1} \end{array}$$

### 3.1. Generalized Eigenvalue Decomposition of Compliance Matrix

For a compliant mechanism, its DOF space and constrained space can be determined by the eigenwrench and eigentwist decomposition of its compliance matrix [***C***] [45], which can be decomposed as follows:(20)$$C\hat{W}=c_{\delta }T_{1}\hat{W}$$
(21)$$CT_{1}\hat{T}=c_{\theta }\hat{T}$$
where $T_{1}=\left(\begin{array}{cc} 0 & 0\\ I & 0 \end{array}\right)$. ***I*** represents the 3×3 identity matrix, 0 is a zero matrix of 3×3, and $c_{\delta }$ and $c_{\theta }$ are called the translational and rotational eigencompliances, respectively. Equations (20) and (21) can be derived from two constraint minimization problems.

The two generalized eigenvalue problems 20 and 21 can be solved with linear algebra. It is well known that each problem yields up to three nonzero eigenvalues $c_{\delta i}$ and $c_{\delta i}$. Their corresponding eigenvectors are called the eigenwrench ${\hat{W}}_{\delta i}$ and eigentwist ${\hat{T}}_{\delta i}$, which are written as:(22)$$\begin{array}{ccc} {\hat{W}}_{\delta i}=\left(\begin{array}{c} f_{i}\\ m_{i} \end{array}\right), & {\hat{T}}_{\delta i}=T_{1}{\hat{W}}_{\delta i}=\left(\begin{array}{c} 0\\ f_{i} \end{array}\right), & \left(i=1,2,3\right) \end{array}$$
(23)$$\begin{array}{ccc} {\hat{T}}_{\theta i}=\left(\begin{array}{c} \theta_{i}\\ \delta_{i} \end{array}\right), & {\hat{W}}_{\theta i}=T_{1}{\hat{T}}_{\theta i}=\left(\begin{array}{c} 0\\ \theta_{i} \end{array}\right), & \left(i=1,2,3\right) \end{array}$$

The above two generalized eigenvalue decompositions can be summarized as the following equation:(24)$$\begin{array}{c} C\left(\begin{array}{cccccc} {\hat{W}}_{\delta 1} & {\hat{W}}_{\delta 2} & {\hat{W}}_{\delta 3} & {\hat{W}}_{\theta 1} & {\hat{W}}_{\theta 2} & {\hat{W}}_{\theta 3} \end{array}\right)=\\ \left(\begin{array}{cccccc} {\hat{T}}_{\delta 1} & {\hat{T}}_{\delta 2} & {\hat{T}}_{\delta 3} & {\hat{T}}_{\theta 1} & {\hat{T}}_{\theta 2} & {\hat{T}}_{\theta 3} \end{array}\right)\left(\begin{array}{cccccc} c_{\delta 1} & & & & & \\ & c_{\delta 2} & & & & \\ & & c_{\delta 3} & & & \\ & & & c_{\theta 1} & & \\ & & & & c_{\theta 2} & \\ & & & & & c_{\theta 3} \end{array}\right) \end{array}$$
where ${\hat{W}}_{\delta i}$ and ${\hat{T}}_{\delta i}$ are the wrench–compliance axes and twist–compliance axes respectively. Physically, the six eigencompliances $c_{\delta i}$ and $c_{\delta i}$ are the magnitudes of translation and rotation along these compliance axes assuming that the directional vector of these axes are normalized.

### 3.2. Judging the DOF of the CPM by the Symbolic Formulation

The compliance matrix for the novel CPM can be obtained by $C=K^{-1}$, which can be written as:(25)$$C=K^{-1}=\left(\begin{array}{cccccc} 0 & 0 & -1.2621\times 10^{-4} & 4.2738\times 10^{-7} & 0 & 0\\ 0 & 0 & 0 & 0 & 4.5732\times 10^{-6} & 0\\ 1.2621\times 10^{-4} & 0 & 0 & 0 & 0 & 4.2738\times 10^{-7}\\ 0.0374 & 0 & 0 & 0 & 0 & 1.2621\times 10^{-4}\\ 0 & 2.4464\times 10^{-6} & 0 & 0 & 0 & 0\\ 0 & 0 & 0.0374 & -1.2621\times 10^{-4} & 0 & 0 \end{array}\right)$$
where the units of the upper left, upper right, and lower left 3 × 3 blocks, are rad/N, rad/Nmm and mm/N, respectively.

According to Equations (20) and (21), the translational eigencompliances and rotational eigencompliances can be expressed as:(26)$$\begin{array}{c} c_{\delta }=\left(1.0818\times 10^{-4},2.4464\times 10^{-6},1.0818\times 10^{-4}\right)\;\mathrm{mm}/N\\ c_{\theta }=\left(4.2738\times 10^{-7},4.5732\times 10^{-2},4.2738\times 10^{-7}\right)\;\mathrm{rad}/\mathrm{Nmm} \end{array}$$

According to Equation (26), since the units are inconsistent, the DOF of the mechanism cannot be directly determined. A characteristic length should be selected in order to determine the DOF. Since the mechanism is a closed-chain type, selection criterion **1** is used to select the characteristic length value $l_{c}=r_{1}=72.55\;\mathrm{mm}$. According to Reference [38], the characteristic length *l*_c_ only affects the rotational eigencompliances, instead of the translational eigencompliances, ${\tilde{c}}_{\delta }=c_{\delta },\;{\tilde{c}}_{\theta }=l_{c}^{2}c_{\delta }$. The rotation eigencompliances in Equation (26) are multiplied by $l_{c}^{2}$. Then the translational eigencompliances and rotational eigencompliances are respectively expressed as follows:(27)$$\begin{array}{c} {\tilde{c}}_{\delta }=\left(1.0818\times 10^{-4},\;2.4464\times 10^{-6},\;1.0818\times 10^{-4}\right)\;\mathrm{mm}/N\\ {\tilde{c}}_{\theta }=\left(2.1944\times 10^{-3},\;2.3481\times 10^{-2},\;2.1944\times 10^{-3}\right)\;\mathrm{mm}/N \end{array}$$

According to Equation (27), the eigencompliance values are listed in ascending order as follows:(28)$$\begin{array}{c} \left(c_{1}=2.3481\times 10^{-2}\right)>\left(c_{2}=2.1944\times 10^{-3}\right)\geq \left(c_{3}=2.1944\times 10^{-3}\right)>\\ \left(c_{4}=1.0818\times 10^{-4}\right)\geq \left(c_{5}=1.0818\times 10^{-4}\right)>\left(c_{6}=2.4464\times 10^{-6}\right) \end{array}$$

According to Equation (28), the calculation of the compliance ratio can be expressed as:(29)$$\begin{array}{c} CR_{1}=1,\;CR_{2}=\;9.3455\times \;10^{-2},\;CR_{3}=\;9.3455\times \;10^{-2}\\ CR_{4}=\;4.6071\times \;10^{-3},\;CR_{5}=\;4.6071\times \;10^{-3},\;CR_{6}=\;1.0419\times \;10^{-4} \end{array}$$

According to Equation (29), if the threshold is chosen to be *ε* = 0.001, obviously there are *m* relatively small eigencompliances, that is *m* = 1. According to DOF criterion 1, the DOF of the compliance mechanism is *N* = 6 − *m* = 5.

According to Equations (22) and (23), the eigenwrench and eigentwist of the mechanism can be written as:(30)$$\left(\begin{array}{cc} {\hat{W}}_{\delta } & {\hat{W}}_{\theta } \end{array}\right)=\left(\begin{array}{cccccc} 1 & 0 & 0 & 0 & 0 & 0\\ 0 & 1 & 0 & 0 & 0 & 0\\ 0 & 0 & 1 & 0 & 0 & 0\\ 0 & 0 & 295.3095 & 1 & 0 & 0.0044\\ 0 & 0 & 0 & 0 & 1 & 0\\ -295.3095 & 1 & -0.0082 & 228.4822 & 0 & 1 \end{array}\right)$$
(31)$$\left(\begin{array}{cc} {\hat{T}}_{\delta } & {\hat{T}}_{\theta } \end{array}\right)=\left(\begin{array}{cccccc} 0 & 0 & 0 & 1 & 0 & 0.0044\\ 0 & 0 & 0 & 0 & 1 & 0\\ 0 & 0 & 0 & 227.8769 & 0 & 1\\ 1 & 0 & 0 & 6.7474\times 10^{4} & 0 & 296.1\\ 0 & 1 & 0 & 0 & 0 & 0\\ 0 & 0 & 1 & -296.2400 & 0 & -1.300 \end{array}\right)$$
where the eigentwists represent the motion of the compliant mechanism when exerted to the eigenwrenches. If the eigencompliance is larger, the corresponding eigentwist represents the mobility of the flexure mechanism along the direction. The direction depends on the vector *θ* of the eigentwist when the normal value of *θ* is not equal to zero and otherwise depends on the vector *δ*. As for the CPM shown in Figure 5a, it can be seen from Equation (31) that the eigentwist corresponding to the five larger eigencompliances are ${\hat{T}}_{\delta i}$ and ${\hat{T}}_{\theta i}$, so the mechanism has five DOFs: three rotational DOFs about the *x*, *y* and *z*-axes, and two translational DOFs about *x* and *z*-axes. According to the screw theory,$\overset{⌢}{T}=\left(S|S^{0}\right)=\left(S|c\times S+pS\right)$, where vector ***S*** denotes the direction of the twist line and is independent of origin; the vector ***S***^0^ is origin-dependent. ***c*** denotes the radius vector of the twist line to the origin of the global coordinate system, and p is called the pitch of a screw given by $p=\frac{S\cdot S^{0}}{S\cdot S}$ and $c=\frac{S\times S^{0}}{S\cdot S}$. The vector $c={\left(\begin{array}{ccc} 0 & 0 & 0 \end{array}\right)}^{T}$ is located at the origin of the global coordinate, so its translation along and rotation around the global coordinate system *x*, *y* and *z*-axes can be expressed as: ${\overset{⌢}{T}}_{\delta x}={\left(\begin{array}{cccccc} 0 & 0 & 0 & 1 & 0 & 0 \end{array}\right)}^{T}$, ${\overset{⌢}{T}}_{\delta y}={\left(\begin{array}{cccccc} 0 & 0 & 0 & 0 & 1 & 0 \end{array}\right)}^{T}$, ${\overset{⌢}{T}}_{\delta z}={\left(\begin{array}{cccccc} 0 & 0 & 0 & 0 & 0 & 1 \end{array}\right)}^{T}$, ${\overset{⌢}{T}}_{\theta x}={\left(\begin{array}{cccccc} 1 & 0 & 0 & 0 & 0 & 0 \end{array}\right)}^{T}$, ${\overset{⌢}{T}}_{\theta y}={\left(\begin{array}{cccccc} 0 & 1 & 0 & 0 & 0 & 0 \end{array}\right)}^{T}$, ${\overset{⌢}{T}}_{\theta z}={\left(\begin{array}{cccccc} 0 & 0 & 1 & 0 & 0 & 0 \end{array}\right)}^{T}$.

For example, ${\overset{⌢}{T}}_{\delta x}={\left(\begin{array}{cccccc} 0 & 0 & 0 & 1 & 0 & 0 \end{array}\right)}^{T}$ denotes the translation along the *x*-axis of the global coordinate system. By calculation, p = 0 and $c={\left(\begin{array}{ccc} 0 & 0 & 0 \end{array}\right)}^{T}$ can be obtained which denotes the pitch of a screw and is located at the origin of the global coordinate, respectively. As for ${\overset{⌢}{T}}_{\delta y}={\left(\begin{array}{cccccc} 0 & 0 & 0 & 0 & 1 & 0 \end{array}\right)}^{T}$, since $CR_{6}=1.0419\times 10^{-4}<\varepsilon$, there is no DOF along the *y*-axis of the global coordinate system.

Moreover, ${\overset{⌢}{T}}_{\theta z}={\left(\begin{array}{cccccc} 0.0044 & 0 & 1 & -296.1 & 0 & -1.3 \end{array}\right)}^{T}$ denotes the rotation around the *z*-axis of the global coordinate system. By calculation, p = 1 and $c={\left(\begin{array}{ccc} -296.1 & 0 & -2.3 \end{array}\right)}^{T}$ can be obtained, which denotes the pitch of a screw and the radius vector of the twist line to the origin of the global coordinate system, respectively.

Since each branch of the parallel mechanism does not yet include an actuating unit, there is one less DOF for the mobile platform along the *y*-axis. After adding the actuating unit, according to the parallel robot actuating mode and the symmetry of the parallel mechanism, the mobile platform can perform the displacement in the *y*-axis.

The characteristic length *l*_c_ = *r*_1_ = 72.55 mm selected according to the selection criterion 1 in Reference [40] is not an exact value and is selected according to the simplified formula. The value of the characteristic length does not need high accuracy, and the characteristic length value within a certain range does not affect the judgment of the number DOF of the mechanism based on the symbolic formulation method, as shown in Figure 6.

## 4. Numerical and Finite Element Method Normal Variable Analysis of CPM

Two methods are used to obtain the deformation data of the CPM, one is the numerical method by applying a certain load, and the other is the finite element simulation with commercial software. The deformation data obtained by the two methods were analyzed and compared in this section. Since the CPM can achieve a high-precision motion, the load is static or quasi-static with a low speed, and the mobile platform is very slow, so there is no need to analyze other performances such as velocity and acceleration, and this paper mainly considers its static stiffness. One can transform the coordinate matrix transformation based on Equation (32):(32)$$K=\sum_{i=1}^{6}\left[Ad_{i}\right]\left[K_{i}\right]{\left[Ad_{i}\right]}^{-1}$$

As shown in Equation (32), ***K*** is the stiffness matrix of the mechanism obtained by the coordinate transformation method. According to Equation (12), the stiffness matrix can be inversely transformed to obtain the compliance matrix of the mechanism.

Applying a load matrix ***CW*** to the origin of the fixed coordinate system at the center of the upper surface of the mobile platform, the deformation of the compliant mechanism can be obtained numerically, as follows:(33)$$W={\left(\begin{array}{cccccc} F_{x} & F_{y} & F_{z} & M_{x} & M_{y} & M_{z} \end{array}\right)}^{T}$$
(34)T=CW

For the pure force along the x-axis, the deformation is mainly the translation along the x-axis. The translation and rotation on the other axes are so small that they can be ignored. Therefore, for pure force, the amount of translation along the *x*-axis direction is mainly acquired.

In this paper, the material of the compliant spherical joint is high-quality carbon structural steel 60, with high strength, hardness and elasticity, and it has a tensile strength of $\sigma_{b}\geq 675\;\mathrm{MPa}$. The basic configuration of the finite element static analysis is consistent with the compliant spherical joint; According to the FEA result, when the force applied to the mobile platform along the *x*-axis reaches 32.59 N, the tension of the compliant spherical joint will exceed 675 MPa. Therefore, within the range of yield strength of the selected material, by applying a force in the range of 0 N ~ 32.59 N along the *x*-axis to the mobile platform, a total of 33 sets of numerical and finite element methods are used to obtain the deformation of the mobile platform, as shown in Figure 7.

In Figure 7, Δ*δ_x_* denotes the displacement deformation along the *x*-axis and *μ* represents the displacement deformation difference obtained by the two methods along the *x*-axis.

As shown in Figure 7a, in the range of yield strength, the magnitude of the force acting on the mobile platform in the *x*-direction is proportional to the displacement deformation of the mobile platform along this direction. It can be seen from Figure 7b, that the larger the force is, the larger the error between the numerical method and the finite element method is; moreover, the error increases linearly, and the maximum error of the analytical method relative to the numerical method is 6.89%.

The above data is an example of displacement deformation caused by applying a force along the *x*-axis to the mobile platform. The force applied to the *z*-axis and the torque around the *x*, *y*, and *z*-axes will correspondingly produce a displacement and angular deformation. After the branches are installed with the driving, according to the parallel robot driving mode and the symmetry of the parallel mechanism, the mobile platform can perform the *y*-direction displacement movement along the *y*-axis.

Under the load condition, the displacement obtained by the numerical method and analytical method has some deviations on some data points, but the error is less than 6.89% relative to the numerical method, which is within the acceptable range. Therefore, the results of the numerical and finite element methods on the displacement and angular deformation of the six DOFs parallel mechanism under load conditions agree with each other and satisfy the linear relationship.

Additionally, the load capacity of the CPM was analyzed by the FEA method in this paper. As shown in Figure 8, under the condition of existing gravity, a load was applied vertically downward along the *y* coordinate axis to the origin of the fixed coordinate system at the center of the mobile platform in the CPM. When the load was fixed, the coordinate system reached 3826.82 N; the equivalent stress of each component of the mechanism reached the tensile strength limit of 675 MPa for 60 steel materials at each compliant spherical joint. However, the load capacity of the CPM was weakened under the non-initial position, because after the motion of the mobile platform, the compliant spherical joint was not only subjected to the axial force but also to the bending moment. According to the finite element analysis, when the mobile platform moves 1.20 mm along the *x* or *z*-axis, the CPM mobile platform can withstand 733.82 N without exceeding the material elastic limit. According to the finite element simulation, the closer to the boundary of the workspace of the CPM, the smaller the bearing capacity of the CPM, and the bearing capacity can be maximized at its initial position. It can be seen that the CPM has a certain ability to withstand large loads in its workspace. This static simulation thus shows that the CPM is able to bear large loads.

Table 2 shows the corresponding deformation values obtained by finite element simulation analysis of some force/torque values taken randomly within the linear elastic range of the mechanism material. The deformation values measured in the table are the value outputs by the finite element software.

## 5. Conclusions

A novel CPM able to effectively improve the load capacity was designed by employing a force-inversed concept of the compliant spherical joint in this paper. This type of mechanism is recommended for high-precision, multi-degree-of-freedom and large-load applications, and is widely applied in sensors andassembly of micro-systems. The DOF was analyzed with the screw theory and symbolic formulation method. The DOF and structural stiffness were verified by finite element simulation. Some meaningful conclusions can be drawn as follows.

(1)The existing six DOFs CPM mobile platforms fail to bear a considerable load because of the bucking of the flexural joints. However, the proposed novel mechanism will be able to bear large loads, as much as a 32.59 N force along the *x* and *z*-axes.(2)The DOF of the mechanism was numerically analyzed by the generalized eigenvalue and eigenvector decomposition of the compliance matrix for the compliance matrix symbolic formulation of the whole mechanism, and the number of the DOF is verified by the numerical and finite element method. Under the same load, the deformation error obtained by the two methods is less than 6.89% compared with the finite element method; in addition, the load and deformation increases linearly within the yield strength range of the material.

## Figures and Tables

**Figure 1 sensors-19-00828-f001:**
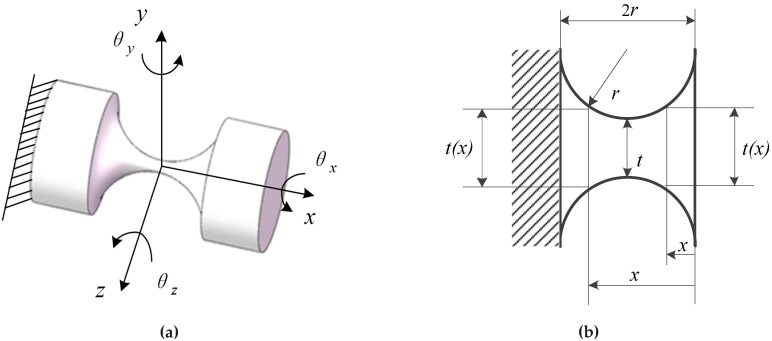
The model and geometry of an RCCS compliant spherical joint: (**a**) model of an RCCS; (**b**) geometry of an RCCS compliant spherical joint.

**Figure 2 sensors-19-00828-f002:**
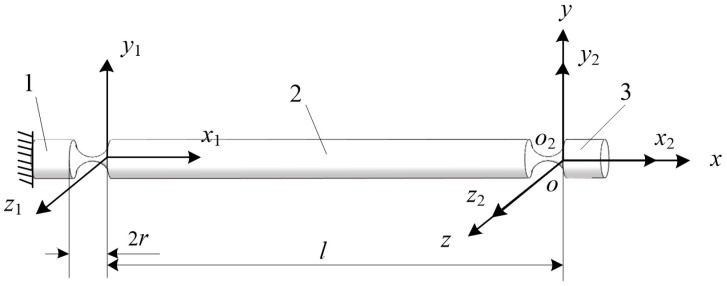
A serial chain of two RCCS compliant spherical joints.

**Figure 3 sensors-19-00828-f003:**
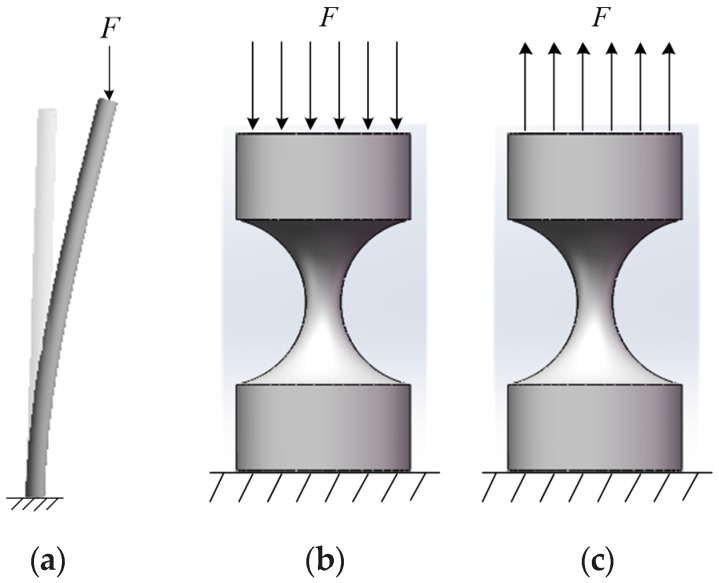
Buckling phenomenon of the slender rod: (**a**) the slender rod buckling phenomenon; (**b**) pressure on an RCCS flexural joints and (**c**) pull on RCCS flexural joints.

**Figure 4 sensors-19-00828-f004:**
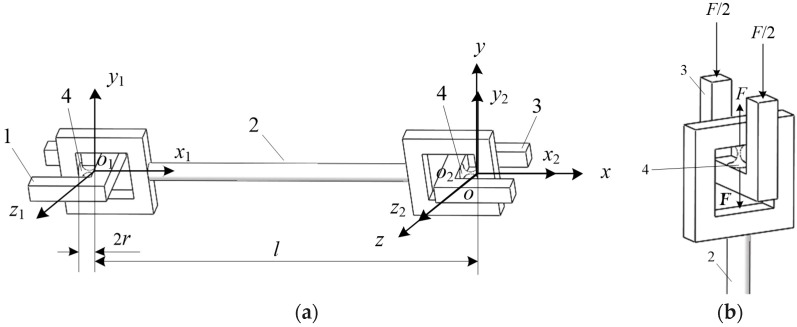
A novel serial RCCS flexural joint chain structure based on an RCCS flexural joint: (**a**) a serial chain of two compliant spherical joints with the force-inverted structure; (**b**) compliant spherical joint force-inverted structure.

**Figure 5 sensors-19-00828-f005:**
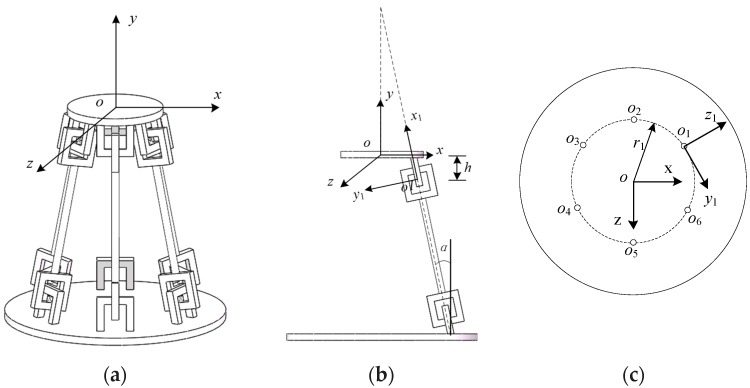
Views of the CPM: (**a**) three-dimensional model of the CPM; (**b**) left view of the CPM; (**c**) top view of the CPM.

**Figure 6 sensors-19-00828-f006:**
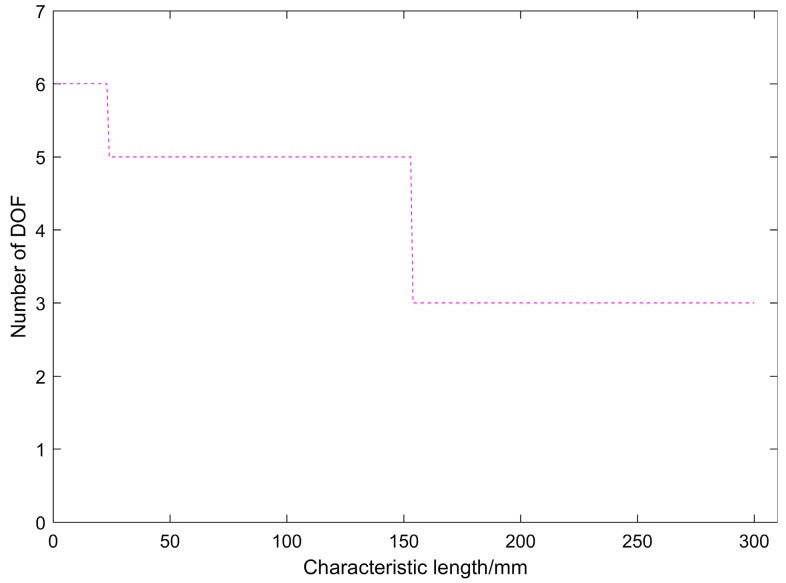
The relationship between the number of DOF and the value of the characteristic length.

**Figure 7 sensors-19-00828-f007:**
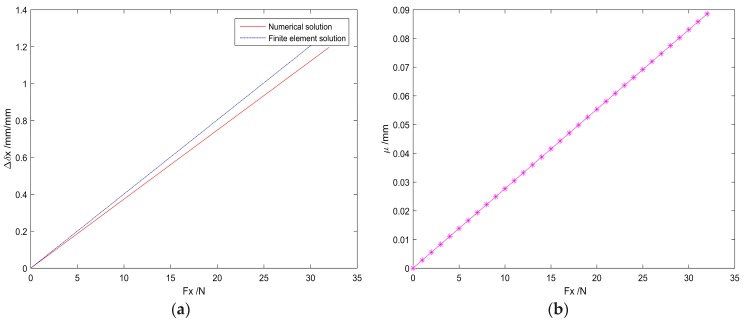
Deformation of the mobile platform that is obtained by analytical and numerical methods: (**a**) displacement along the *x*-axis under axis under load; (**b**) displacement difference along the *x* load.

**Figure 8 sensors-19-00828-f008:**
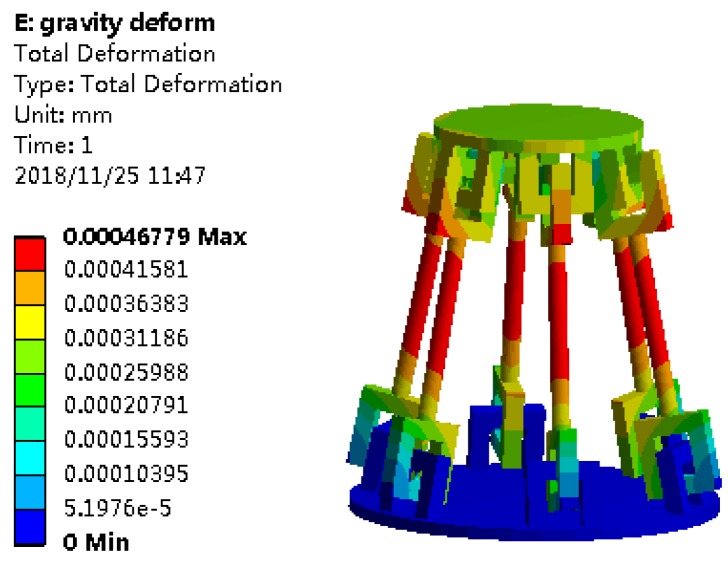
The load nephograms of the CPM with bearing gravity.

**Table 1 sensors-19-00828-t001:** Parameters of the 6-DOF CPM.

Variable	Value	Unit
*E*	200	GPa
*ν*	0.3	mm
*t*	2	mm
*r*	5	mm
*l*	240	mm
*r* _1_	72.55	mm
*α* _1_	0	°
*α* _2_	0	°
*α* _3_	0	°
*α* _4_	0	°
*α* _5_	0	°
*α* _6_	0	°
*β* _1_	30	°
*β* _2_	90	°
*β* _3_	150	°
*β* _4_	210	°
*β* _5_	270	°
*β* _6_	330	°
*γ* _1_	102	°
*γ* _2_	102	°
*γ* _3_	102	°
*γ* _4_	102	°
*γ* _5_	102	°
*γ* _6_	102	°
*ρ* _1_	30	°
*ρ* _2_	90	°
*ρ* _3_	150	°
*ρ* _4_	210	°
*ρ* _5_	270	°
*ρ* _6_	330	°
*ζ* _1_	150	°
*ζ* _2_	180	°
*ζ* _3_	210	°
*ζ* _4_	240	°
*ζ* _5_	270	°
*ζ* _6_	300	°
*h*	35.78	mm

**Table 2 sensors-19-00828-t002:** Partial force/torque values corresponding to deformation values.

*F_z_* (N)	Δ*δ_z_* (mm)	*M_x_* (Nmm)	Δ*θ_x_* (°)	*M_y_* (Nmm)	Δ*θ_y_* (°)	*M_z_* (Nmm)	Δ*θ_z_* (°)
5.54	0.22	5536.68	0.17	1474.42	0.44	4281.42	0.13
9.91	0.40	6619.62	0.20	1539.20	0.46	4958.33	0.15
14.09	0.57	6876.59	0.21	1675.83	0.50	5547.47	0.17
14.45	0.57	7302.79	0.22	1915.12	0.57	5839.59	0.18
17.79	0.71	7394.97	0.23	1945.70	0.58	6254.18	0.19
22.21	0.89	8110.67	0.25	2396.66	0.72	7216.39	0.22
24.09	0.97	8962.77	0.27	2435.86	0.73	7534.12	0.23
25.71	1.03	9247.49	0.28	2667.90	0.80	7888.29	0.24
26.44	1.06	9405.11	0.29	2966.74	0.90	8155.98	0.25
29.62	1.03	9751.75	0.30	3047.80	0.91	8976.38	0.27
31.03	1.25	10,003.81	0.31	3070.20	0.92	9150.62	0.28
32.34	1.30	10,125.49	0.31	3593.10	1.08	9909.63	0.30

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
