# Peer review of "Design and DOF Analysis of a Novel Compliant Parallel Mechanism for Large Load"

_sensors, 2019, doi:10.3390/s19040828_

Round 1

Reviewer 1 Report

The submitted paper presents the analysis and investigation of a universal joint based on force-inversion and of a compliant parallel mechanism (mobile platform) consisting of 12 of these universal force-inversed hinges. The investigations are made under special consideration of two important compliant mechanisms aspects, the DOF and the load capacity. Especially the latter aspect has been insufficiently investigated so far. Thus, the paper seems to be of highly relevant content. The hinge and the mechanism results for the DOF and structural stiffness are obtained based on the screw theory and symbolic formulation method. Finally, the analytical results are verified by means of FEM simulations.  

At the first impression the paper seems to be well prepared. But while reading in detail too much small and big questions arise which must be answered and explained through a major revision before the paper can be accepted for publication. For example, the choice of the hinge shape and parameters, the typical hinge stroke/rotation angle, the influence of the not perfectly rigid link segments on the high-precision output motion, the influence of geometrical non-linearities and their possible consideration, the choice of the geometric parameters, and the FEM settings. But most important is, that there is no explanation of the relation of the paper’s content/focus to the journal “Sensors”. There is no hint and description in the text why you decide submitting for this journal. In addition, there are several Typos like missing spaces in the equations or before units, grammar things, and so on.

The following concrete things should be revised please:

Line 18/19: Please check the expression of this sentence, there might be something wrong with the grammar (something is missing).

Line 24: What is the difference between “movement” and “motion” and why do you use “movement” here?

Line 25: Better to use the term “rigid-body mechanism”.

Line 35: “Structural” parameters is used not consistently because structure describes material and geometry. Better to use “geometric” parameters here.

Line 78: Why do you use the term ”Right Circular Cross-Section Corner-Filleted) flexural joint”? From Fig. 2 and also Eq. 2 it seems that a right-semi-circular flexure hinge with a variable circular cross-section is used. Where are the corner-fillets? Why do you choose this semi-circular shape? From notch flexure hinges with rectangular cross-section it is known, that the circular shape is not optimal and that the hinge performance can be improved by using special shapes. Perhaps there would be possible and of benefit for universal hinges too, like studied in reference [36]? Please study and refer to the following references for example in this context.

(a) Zelenika, S.; Munteanu, M. G.; Bona, F. De (2009): Optimized flexural hinge shapes for microsystems and high-precision applications. In: Mechanism and Machine Theory 44 (10), pp. 1826–1839. DOI: 10.1016/j.mechmachtheory.2009.03.007.

(b) Meng, Q., Berselli, G., Vertechy, R., and Castelli, V. P.: An Improved Method for Designing Flexure-Based Nonlinear Springs, in: Proceedings of the ASME 2012 36th Mechanisms and Robotics Conference, ASME 2012 36th Mechanisms and Robotics Conference, Chicago, Illinois, USA, August 12–15, pp. 211–219, 2012.

(c) Linß, S.; Schorr, P.; Zentner, L. (2017): General design equations for the rotational stiffness, maximal angular deflection and rotational precision of various notch flexure hinges. In: Mech. Sci. 8 (1), pp. 29–49. DOI: 10.5194/ms-8-29-2017.

(d) Liu, M.; Zhan, J.; Zhu, B.; Zhang, X. (2018): Topology Optimization of Flexure Hinges with Distributed Stress for Flexure-Based Mechanisms. In: 2018 International Conference on Manipulation, Automation and Robotics at Small Scales (MARSS). 2018. Nagoya, Japan, 4/7/2018 - 8/7/2018: IEEE, pp. 1–5.

Line 85: The mentioned three DOFs are the “desired” DOFs only, this may be considered.

Line 88/Fig. 1: Where are the forces and moments applied? Please mention the boundary conditions too. What forces are regarded, follower or direction-constant forces? Are the adjacent link segments considered in the analytical investigation (like shown in Fig. 1a) or not (Fig. 1b)?

Line 95: “where” should be written with a capital “W”.

Line 102: Are the links/bodies model ideally rigid or is the deformation possible too? What could be the influence of this little deformation regarding precision engineering applications? Are large deflections/non-linear theory considered and what is the influence?

Line 125: A space is missing before the bracket.

Line 126/Fig. 4: The buckling phenomenon is clear, but what is the relation to your investigated mechanism and why it is important to consider it here?

Line 135/140: The idea of force inversion within compliant joints is not really new (e.g. Guérinot, A. E.; Magleby, S. P.; Howell, L. L.; Todd, R. H. (2005): Compliant Joint Design Principles for High Compressive Load Situations. In: Journal of mechanical design 127 (4), pp. 774–781). What is the new approach and contribution of your proposed structure?

Line 150: Layout Typos, e.g. l = 130 mm is to be written right (l in italics, and the rest not, and with consideration of spaces).

Line 156: Spaces are missing.

Line 179/Table 1: Why these material and geometric parameters are specified, based on a design of experiments or existing conclusions? 2 mm for the minimum thickness seems to be very thick for a flexural joint, why it is chosen so? Can we still speak about a rotation with an approximately fixed rotational axis and what is the influence of the rotational axis shift on your mechanism properties and the motion accuracy in general? What are the typical/maximum rotation angles of all hinges and are they equal? Fore what stand the given angles alpha, betta and gamma stand for? Please mention them in the figure.

Line 191: Please precise “small rotations” and how large are the relative rotation angles of the flexure hinges again? Why it is not necessary to consider geometrical nonlinearity, as it is known that these are relevant for high-precision compliant mechanisms.

Line 235: Spaces are missing within the formula.

Line 263: You investigate the static stiffness. But what is with the influence of the gravity?

Line 257: What is meant with the numerical solution, the FEM simulation results or are the analytical results obtained by numerical solving?

Line 274: In Section 4 all relevant settings of the FEM simulation are missing and should be briefly described please. That means, what software, what analysis (static structural), concrete loads (load acting point/geometry, directional or non-directional force) and boundary conditions, load-steps, mesh, and element type (2D/3D; with or without middle nodes) do you use? Do you use a linear or nonlinear material model? And most important to flexure hinges when investigating the precision of motion, do you use linear or nonlinear geometry setting? Is the gravity considered? And when analyzing stress, how is the mesh quality and what stress is calculated (mean stress or equivalent stress)? How is the bearing load applied (on a point, edge or face)?

Line 281: What means delta and alpha?

Fig. 7: How is delta calculated?

Line 289: What means “basically identical”?

Line 292:  Are four digits after the comma really necessary for the value of the relative deviation? Is this the maximum error for all simulation and cases?

Line 307: Are four digits after the comma really necessary for the force value?

Fig. 8: Is “nephrogram” a common word, what is meant with this? The deformation of the gravity is in the range of 0.5 µm. This is a very low absolute value, which result depends on lots of FEM settings, like mesh quality, geometric non-linearities, and convergence criteria. Did you check all these settings influences? Is the gravity considered together with the bearing load? From Fig. 8b you couldn’t see anything, because the figure resolution is too poor and the detail too small. What stress is analyzed when comparing the tensile strength? What is with the bending stress in this case?

Line 319: There is a Typo for the word buckling. What is the relation of the presented investigations to the critical buckling load, or how is it possible to have this phenomenon when using the force-inversion hinges? I thought this was the aim, to avoid buckling?

Line 320: What means large loads, compared to what? You may introduce a relation of the possible bearing load and the overall platform weight or the working stroke? What stroke of the hinges and the platform is possible? What weight is related to the given load force?

Line 324: The word “error” is missing perhaps before “obtained”.

Line 327: In the conclusions some result regarding the sensor application is necessary. How can the designer use the results for what sensor?

Line 418: The word “mechanisms” is missing in the reference title.

Author Response

This is my reply, thank you!

Reviewer 2 Report

The authors presented a Novel Compliant 3 Parallel Mechanism for Large Load. 

The design seems convincing only for static loads while it can lead to instability in case of tensile or dynamic loads. The description of the paper is confused because the authors often redefine previously defined variables: 

Same Ci in eq.(3)

θ and φ in eq.(9)

θ already defined in Fig. 1b

σ in Table 1 (never defined)

α used both for the displacement deformation around the z-axis and for the angle of the legs

Even the notation of the equations is confusing or imprecise:

Eq.(1), check for the entries of C

Line 114: R1=R2 (not defined)

fix Rx, Ry and Rz in eq.(10)

T1 is too similar to the twist T in eqs.(20)-(21)

The choice of a particular value for the "characteristic length" should be motivated and argued. The results provided in Subsection 3.2 are unclear. Please, provide a better explanation of the eigentwists obtained in eq.(31). It is necessary to verify how these results change by changing the characteristic length.

The results provided in Section 4 are only partial and should be extended to allow the reader to understand the problem and verify the proposed design. Please insert a table with the missing simulation results (force applied to the z-axis and the torques around the x, y, and z-axes).
.

Author Response

This is my reply, thank you!

Round 2

Reviewer 1 Report

The authors have amended the paper according to most of my review comments, thus most provided responses are satisfactory. But there are still some comments which require a Minor Revision before the paper can be published.

Comment 1: There are still lots of Typos in the paper, like missing spaces in the equations (in the text paragraphs), before and after the equal sign, before units, before and after brackets… For a good quality paper this must be corrected in the whole paper again please.

Comment 2, Line 18: What means “normal size”? Is the mechanism calculated and investigated in macro size and later scaled down?

Comment 3, Line 47: There is a Typo before the bracket [13], the point must be deleted.

Comment 4, Line 83: I’m still not convinced that “Right Circular Cross-Section Corner-Filleted” flexural joint is the right term, as I don’t understand the authors response in this case because I cannot see the corner fillets.

Comment 5, Line 70/71: The word “position” is used twice, is this correct?

Comment 6, Line 101: What is a “material mechanism”?

Comment 7, Line 101 and following: Please add a comment or precise the text, that linear geometry setting is used (I expect that this is meant with “linear analysis module) and why it is possible (as you give the reason in the author comment). Please add a reference that it is sufficient in your case for high-precision application, because my experiences are different. However, I think in this context, the non-linear geometry advice is understood wrong, since it has nothing to do with a non-linear material model and exceeding the admissible stress value. What I meant was the non-linear geometry setting what can be realized in Ansys with “nlgeom=on”. In my experience this is necessary to consider when simulating the motion behavior of high-precision compliant mechanism, even when the rotation angle of the hinge is in the range of around 1°.

Comment 8, Line 205/Table 1: Please explain again, like in your responses, why it is allowed to simulate a comparably thick hinge with 2 mm with the linear geometry setting, when you expect influencing errors in the nanometer range. This is questionable without using the non-linear geometry setting. In addition, it is known from the literature, that the thicker the hinge center section is, the higher is the axis shift (which I expect in the micrometer range in your case and not as you response in the nanometer range).

Comment 9, Line 284: Why there is a frame around Fig. 7?

Comment 10, Line 350: The Fig. 9b and the detail in the figure is still of a poor resolution and therefore not sufficient for getting some information about the stress distribution in the hinge.

Author Response

This is our responses, thank you !

Reviewer 2 Report

The authors have improved some parts of the paper. However, the core of the method (subsection 3.2) has not yet described with adequate detail.

1) Passing from eq.(26) to (27) it seems that the value of the characteristic length would be different from 72.55 mm (lc = 0.014 mm instead). Even the units for c_theta in eq.(27) should be changed accordingly.

2) Twists in eq.(31) present some high values, much larger than the unity. Citing the authors' words: "The direction depends on the vector θ of the eigentwist when the normal of θ is not equal to zero, and otherwise depends on the vector  δ". From the fourth to the sixth eigentwist of eq.(31) the direction should depend on vector θ but following the components of vector θ it seems that T_theta does not represent three rotational DOFS about x, y and z axes.

3) Table 2 should be improved distinguishing inputs (forces and torques with appropriate units) from outputs (displacements and rotations with appropriate units).

Please, indicate all units inside parenthesis following the notation: F_x (N) ; \delta_z (°) and so on.

Author Response

This is our responses, thank you !

Round 3

Reviewer 2 Report

Thanks for your comments. I suggest to include part of your Response 2 inside the text. It would help the reader to better understand the role and meaning of the eigentwists. Please check for the fourth twist of (31). I obtain: pitch h =  -6.2110e-04 and r = [0;296.0985;0]. It seems a rotation around the z-axis again because the components of S are [1;0;227.8769] and 227.8769>>1. 

Author Response

This is our responses, thank you !
